# Narrative Forewarnings: A Qualitative Analysis of the Themes Preceding Disorganized Speech in Schizophrenia

**DOI:** 10.3390/bs14030212

**Published:** 2024-03-06

**Authors:** Robert J. Bettis, Laura A. Faith, Ashlynn M. Beard, Brailee A. Whan, Ceouna M. Hegwood, Mahogany A. Monette, Evan J. Myers, Imani S. Linton, Bethany L. Leonhardt, Michelle P. Salyers, Kyle S. Minor

**Affiliations:** 1Department of Psychology, Indiana University-Indianapolis, Indianapolis, IN 46202, USA; robert.j.bettis@vumc.org (R.J.B.); ashbeard@iu.edu (A.M.B.); bwhan@bhcare.org (B.A.W.); chegwood@uci.edu (C.M.H.); mmonette@iu.edu (M.A.M.); evjmyers@iu.edu (E.J.M.); imlinton@iu.edu (I.S.L.); mpsalyer@iu.edu (M.P.S.); 2Department of Psychiatry, Vanderbilt University Medical Center, Nashville, TN 37240, USA; 3Department of Psychiatry, Roudebush VA Medical Center, Indianapolis, IN 46202, USA; laura.faith@va.gov; 4Department of Psychology, University of California-Irvine, Irvine, CA 92697, USA; 5Department of Psychiatry, Indiana University School of Medicine, Indianapolis, IN 46202, USA; blleonha@iupui.edu; 6Prevention and Recovery Center for Early Psychosis, Indianapolis, IN 46202, USA

**Keywords:** disorganized speech, disorganized symptoms, thematic analysis, negative affect, schizophrenia

## Abstract

Disorganized speech is a critical barrier to recovery in schizophrenia, with profound negative impacts on one’s ability to engage with the world. Despite the limited efficacy of existing treatments in addressing disorganization, a qualitative analysis of what leads to disorganization in patient narratives has been lacking. This study addresses this gap through inductive thematic analysis of 30 narrative interviews with individuals with schizophrenia, matched based on whether Formal Thought Disorder (FTD) is present. Through this analysis, we identified four core themes (alienation, interpersonal tension, personal benchmarks, and adverse experiences) and eight subthemes. Our findings suggest that disorganization may serve as a protective mechanism against psychological distress and highlight how the severity of FTD influences these themes. Alienation, particularly due to illness-related stigma, emerged more prominently in those with FTD. The themes of personal benchmarks and interpersonal tension pointed towards a heightened sensitivity to social interactions and self-perception among those with schizophrenia. Adverse experiences, encompassing past challenges, suggest a potential link between trauma and symptom exacerbation. Our qualitative analysis of what themes precede disorganized speech has implications for tailoring psychotherapy. By considering an individual’s specific triggers and level of disorganization, therapy may be more effectively targeted to improve recovery-based outcomes. By identifying themes within patient narratives, this study advances our understanding of the qualitative aspects preceding disorganized speech in schizophrenia, paving the way for more personalized and effective recovery-focused interventions.

## 1. Introduction

Our speech reflects our thought processes and inner world. In schizophrenia research, this has been acknowledged since the first conceptualizations of the disorder when Bleuler [1] noted the “loosening of associations” that are often present. Bleuler [1] asserted this fragmentation (i.e., lack of integration) produced incoherent speech and behavior—precursors for what are described today as disorganized symptoms. Bleuler’s early insights laid the foundation for recognizing disorganized speech as the leading behavioral indicator of disorganized symptoms, with later conceptualizations expanding the definition of disorganized speech to include impairments in communication, such as tangentiality, derailment, and incoherence [2]. Disorganized speech is evident across all stages of schizophrenia from subclinical [3,4,5] to early psychosis [6,7,8] to more prolonged states of the disorder [9,10,11]. Its profound impact on functional outcomes—including role functioning, social functioning, and quality of life [12,13]—poses a formidable barrier to recovery in schizophrenia.

A key step in improving recovery outcomes is to deepen our understanding of the underpinnings of disorganized speech. Although there has been substantial progress in identifying how disorganized symptoms are related to quantitative constructs, little is known about factors that precede and possibly lead to instances of disorganization in those with schizophrenia. One promising potential precursor is negative affect. When compared to healthy control groups, a series of studies have shown that affective reactivity—increased disorganized speech in response to induced negative affect compared to a baseline condition—is greater in early-stage psychosis [6] and schizophrenia populations [14,15,16]. Observed correlations between disorganized symptom severity and affective reactivity [17] further illustrate the importance of negative affect, especially when considering that affective reactivity has shown strong associations with poor social outcomes in schizophrenia spectrum studies [6,18]. Taken together, these findings underscore negative affect as a possible antecedent to disorganized speech, as well as highlighting its potential relevance in recovery and rehabilitation.

Inherent limitations of quantitative research (i.e., lack of context, less focus on themes emerging from participant interviews) necessitate a qualitative approach to better understand disorganized speech. Current qualitative insights, primarily from psychotherapy case studies [19,20,21,22], provide valuable perspectives on the personal significance of disorganization for individuals with schizophrenia. However, these studies are limited in their ability to offer a generalized understanding that is applicable to a broader schizophrenia population. Qualitative inquiry including data of “in the moment” instances of disorganization is a helpful avenue to provide context and a richer understanding of disorganization beyond the breadth provided by quantitative inquiry [23]. Inductive thematic analysis using participant narratives that include a large, diverse sample could yield “middle range” theories [24] on the emergence of disorganized speech, which could be more representative and clinically applicable. For example, Hamm and Firmin [19] explained how disorganization is often viewed as a barrier in psychotherapy, resulting in a void of workable material during sessions. Producing theories as to when disorganization emerges is an important step to show how disorganization can be meaningful for clients and potentially treated by clinicians. Thus, inductive thematic analysis could bridge the gap between individual case studies and broader clinical applications by identifying common themes that lead to disorganization. It could also enhance the therapist’s understanding of the meaning behind disorganized symptoms—rather than viewing these symptoms simply as the result of mental illness itself. In turn, this could lead to more effective, client-centered therapies by tailoring sessions to address these common themes.

Building on the need for qualitative analysis, the current study explores the themes preceding disorganized speech in schizophrenia using patient narratives. Previous studies, such as Leonhardt and colleagues [21], have examined how psychotic content, including disorganization, emerged over the course of therapy in one patient with schizophrenia. They observed that psychotic content often arose in response to themes of inadequacy, vulnerability, and therapist challenges. Identifying these themes allowed the therapist to better understand what content may trigger exacerbations in disorganization. However, these findings were limited to a single case and were from the therapist’s point of view; it is likely that different types of content precede disorganization across different clients in different contexts. Our study aimed to expand this current understanding by exploring a wide range of themes in patient narratives across a diverse group, thereby offering a more comprehensive view of the factors that lead to disorganization.

The severity of disorganization clients experience may also influence the themes preceding disorganization. Formal thought disorder (FTD) represents a severe manifestation of disorganization in schizophrenia [25,26,27]. There is evidence that individuals with FTD face added challenges in cognitive and social functioning compared to those without FTD [28,29]; in turn, this impacts their integration into educational and occupational settings and sets up additional barriers to recovery [30]. These findings highlight the importance of differentiating themes preceding disorganized speech based on FTD severity. This could provide further insights for tailoring interventions, thereby enhancing recovery prospects for individuals with varying levels of disorganized symptoms.

The current study represents the first qualitative analysis aimed at identifying themes from patient narratives that precede disorganized speech in schizophrenia. To build therapeutic frameworks for treating disorganization and removing barriers to recovery, it is vital to understand and highlight the personal meanings that may be associated with the emergence of disorganization. The current understanding of disorganization is based on quantitative data, which lacks depth, context, and process factors potentially relevant to *when* disorganization unfolds. Documenting common themes may serve as a template to aid psychotherapists in attuning to disorganization in their own patients. This study is an inductive exploratory qualitative analysis of themes that preceded disorganization throughout 30 narrative interview transcripts. Based on past literature, we expected to find themes related to negative affect; however, we were also interested in exploring other content that emerged. A secondary objective was to assess if themes differed between those with and without FTD, recognizing the potential for varied themes based on disorganization severity. Insights from this study will enrich our understanding of disorganized speech in schizophrenia, yielding potential guidance for researchers and clinicians in addressing a key barrier to recovery in schizophrenia. 

## 2. Methods

### 2.1. Participants and Data Collection

Narrative interview transcripts (*n* = 30) were obtained as a subset of data from a larger research study [31]. All participants were recruited from a midwestern VA Medical Center and had a diagnosis of schizophrenia or schizoaffective disorder. Inclusion criteria for this study involved (a) ages 18–60, (b) no change in medication or outpatient status 30 days prior to testing, (c) ability to give informed consent, (d) no active substance dependence, (e) no documented intellectual disability, and (f) no history of neurological illness or traumatic brain injury which resulted in loss of consciousness greater than five minutes. 

The total sample consisted of all males (*n* = 30, 100%), was predominantly middle-aged (*M* = 43.8, *SD* = 9.72), and contained both Black (*n* = 20, 67%) and White (*n* = 10, 33%) participants. To analyze the secondary aim, participants were further divided into FTD (*n* = 15) and non-FTD (*n* = 15) subgroups. Participants rated ≥4 (“moderate” or greater) on the Positive and Negative Syndrome Scale (PANSS; [32]) conceptual disorganization item composed the FTD group. The FTD and non-FTD groups were matched on age, gender, race, and education using a matched pairs design. In the current study, purposive sampling from the parent study [31] was applied to emphasize theory-based variation [33]. 

### 2.2. Measures

**Narratives:** The Indiana Psychiatric Illness Interview (IPII; [34]) was utilized to collect participant speech data for qualitative analysis. The IPII contains open-ended questions primarily for exploring the participant’s recollection of their life narrative and experience of mental illness. Administrators of the IPII are trained to give minimal prompts or directions aside from asking five selected queries. This offers participants the freedom to respond with minimal outside influence. The IPII usually ranges from 30 to 60 min, depending on the participant’s responses. Interviews were then transcribed by trained research assistants (RAs) and checked by a separate RA to enhance accuracy. The transcripts from this study were derived from the following prompt: “I’d like you to tell me the story of your life, in as much detail as you can, from as early as you can remember up to now. If it helps you to organize your story, you can divide it into chapters or sections. Any questions?”

**Instances of Disorganized Speech within Transcripts:** Two methods were used to identify specific instances of disorganized speech within participant transcripts. First, the Communication Disturbances Index (CDI; [35]) was initially used to rate communication disturbances. All transcripts were rated for communication disturbances in a previous study [36] by two CDI-trained raters: a clinical psychology graduate student and the lead author (both male; White). They identified six different types of communication disturbances (vague references, confused references, missing information references, ambiguous word meanings, wrong word references, structural unclarity) in individual IPII interviews (see [35,37]). The study achieved strong interrater reliability prior to consensus meetings occurring (intraclass *r* = 0.90). While the CDI is typically used to represent the overall amount of disorganized speech, we used it in this study to identify specific instances of disorganization within transcripts.

Second, specific instances of CDI disturbance were further separated into clinically meaningful instances of disorganization based on the conceptual disorganization item definition in the PANSS [32]. The CDI identifies subtle signs of disorganization, and our goal in this study was to determine what themes preceded clinically meaningful instances of disorganization. The lead author and three research assistants (one male, three female; one Black, three White) identified communication disturbances consistent with a 4 (“moderate”) or greater in the conceptual disorganization section of the PANSS (i.e., “able to focus thoughts when communications are brief and structured, but becomes loose or irrelevant when dealing with more complex communications or when under minimal pressure”). All were CDI-trained raters under the supervision of a clinical psychologist and clinical psychology graduate student. Ratings were made independently, and consensus meetings were held to finalize ratings. The lead author was aware of participant FTD groupings prior to rating disorganization, but other raters were blind to all grouping and participant demographic data. 

### 2.3. Qualitative Analyses

This study is an exploratory inductive thematic analysis of the content preceding instances of disorganization. Open-ended narrative interviews, prompted by a single question, were used to provide data most appropriate for our investigation to understand the organic process of disorganization [24,38]. The speech was generated with little to no prompting from the environment or another person, allowing for all potential triggers of disorganization to be seen within the person’s telling of their narrative. 

Analyses began with four qualitative coders independently coding for the themes they interpreted as most significant in the content immediately preceding a highlighted instance of disorganized speech (i.e., an instance where CDI was coded and characterized as “Moderate” or higher using the PANSS conceptual disorganization definition). “Immediately preceding” did not have a strict definition but typically referred to within five lines prior to the instance of disorganization. In rarer cases, coders might have looked between 5 and 10 lines before the instance of disorganization. There were at least two coders per transcript. Codes were organized into themes after patterns were identified from the data (i.e., coded content). All codes were finalized in group consensus meetings including all four coders to ensure themes were accurately representing the data from four perspectives. Consensus meetings were held for approximately every four transcripts coded. For the first 18 transcripts coded, codes were iteratively redefined, added, or removed. This involved continuously contrasting similarities and differences of emergent themes with existing codes until 7–12 codes were developed to represent all emergent phenomena. After the first 18 transcripts, a codebook of 8 codes was developed (Table 1). This codebook accounted for all themes present in the last 12 transcripts, with all coders coding at least 4 of the remaining transcripts. At this point, the codebook was determined to be saturated given its consistency in identifying themes preceding conceptual disorganization. Once the codebook was complete, the 8 codes were divided into 4 categories for further analysis. Themes were also organized into subthemes (see Table 1). We then examined the frequency of how often different types of codes preceded disorganization in FTD and non-FTD subgroups.

Methodological integrity for this study was informed by recommendations from the Task Force on Resources for the Publication of Qualitative Research of the Society for Qualitative Inquiry in Psychology, a section of Division 5 of the American Psychological Association [38]. Given that the data were collected through a previous study [31], the only contact coders had with participants was working with their transcribed interviews. Additionally, 3 of the 4 coders, excluding the lead author, were blind to participant grouping or demographic data prior to coding each transcript. The lead author of this study had a theoretical background acknowledging affective reactivity and metacognitive perspectives of serious mental illness. To balance this perspective, the three coders approached this study without a similar theoretical background. The qualitative analyses were supervised by two clinical psychologists, one of whom was experienced in qualitative work.

## 3. Results

The final codebook (see Table 1) identified four themes: alienation, interpersonal tension, personal benchmarks, and adverse experiences. Within each, there were two subthemes. For both FTD groups, the most frequent topic preceding disorganization was relationship tension. For FTD, the next most common themes were alienation related to illness and social alienation. For the non-FTD group, the next most common themes were past hardship and inadequacy. Notably, alienation related to illness experience occurred almost exclusively in the FTD group. Below we outline each theme and subtheme, provide quotes to characterize the themes/subthemes, and offer examples illustrating how the FTD and non-FTD groups differed in instances that preceded disorganization (also see Table 1). 

**Table 1 behavsci-14-00212-t001:** Code definitions, categories, and FTD vs. non-FTD code distributions.

		Overall Cohort (*n* = 30)		FTD Group (*n* = 15)		Non-FTD Group (*n* = 15)	
Alienation		** *n* **	**Occurrences (%) of Disorganization**	** *n* **	**Occurrences (%) of Disorganization**	** *n* **	**Occurrences (%) of Disorganization**
Alienation Related to Illness Experience	Experiences relating to the lived experience of serious mental illness (e.g., mental illness identity, symptoms, rehabilitation) connected with descriptions of being misunderstood or different.	13	16 (15.7%)	12	15 (19.7%)	1	1 (3.8%)
Alienation Related to Social Experience	Feelings and experiences of being targeted, isolated, not belonging, or unaccepted while in a group of people.	7	16 (15.7%)	4	13 (17.1%)	3	3 (11.5%)
Interpersonal Tension							
Relationship Tension	Vulnerable experiences with others. These could be describing platonic, romantic, or professional connections, an interpersonal dynamic, their perception, or their interpretation of others.	12	21 (20.6%)	8	16 (21.0%)	4	5 (19.2%)
T really does noAggressive Tension	Experiences involving violence or aggression. Could be a victim, perpetrator, or generally violent situation.	7	8 (7.8%)	4	5 (6.6%)	3	3 (11.5%)
Personal Benchmarks							
Recognizing Missed Potential	Recognition of past or present potential in their narrative (e.g., expression of hopes, motives, or past achievements) which transgressed into self-criticism or disappointment.	11	12 (11.8%)	9	9 (11.8%)	2	3 (11.5%)
Inadequacy	Description of experiences (e.g., personality, upbringing, vocation, relationships, resources) often viewed as lacking or insufficient.	9	11 (10.8%)	5	7 (9.2%)	4	4 (15.4%)
Adverse Experience							
Past Hardship	Recollection of negative, painful, or adverse experiences without an overt emotional process.	7	11 (10.8%)	4	6 (7.9%)	3	5 (19.2%)
Grief	Negative emotions which come with significant loss (e.g., loss of another person, resources, security, health, way of being).	6	7 (6.9%)	5	5 (6.6%)	1	2 (7.7%)
TOTAL:			102 (100%)		76 (100%)		26 (100%)
Mean (Standard Deviation)			3.43 (3.69)		5.13 (4.41)		1.73 (1.62)
Range (# with NO Occurrences)			0–16 (4)		1–16 (0)		0–4 (4)

Note: # = number.

In most comparisons between the two groups, a more intimate and detailed description of a given theme was present preceding disorganization in the non-FTD group. It appeared that less negative affect was required to evoke disorganization in the FTD group. This could be seen across several codes through less coherent descriptions of challenging events but was most clear in the alienation codes (which were more frequently found in the FTD group).

### 3.1. Alienation

The theme of alienation emerged as a precursor to disorganized speech, encompassing feelings of estrangement and misunderstanding, especially in relation to mental illness and social experiences. To represent both types of alienation we developed two subthemes: alienation related to illness experience and alienation related to social experience.

#### 3.1.1. Alienation Related to Illness Experience

Participants, particularly those with FTD, often described feeling outcasted by society when they described their mental illness or treatment experiences. In their narratives, participants could frequently initiate a clear description of medication, therapy, or hospitalization. However, these descriptions typically contained negative affective descriptions, which would precede an instance of disorganization. The following quote demonstrates such phenomena well: 

FTD Participant: “They put me on bad medication. I was having side effects from bad medication. They re-hospitalized me at [hospital]. I sat there for 6 months. Even though I felt like a prisoner of war, I was still fighting other people animals”.

After mentioning medication and hospitalization, this participant’s speech becomes disorganized when saying “still fighting other people animals”, without further elaboration or context. The disorganization may not have emerged from simply poor treatment experiences but the wider social implications. Taking a deeper look, hospitalization may have felt dehumanizing and war-like to this individual when he was experiencing medication side effects.

Acknowledgment of treatment could also be subtle, as this participant referred to the VA medical center when mentioning “here”.

FTD Participant: “So I’m doing pretty good here and everything … up to now, I feel pretty good and stuff and dealing with the frustration of not smoking helps out quite a bit too. Even though you might not be sitting down in one of the chairs in the clinic or anything, probably, if you were in the hospital or in or out or something … something like that and uh …”.

Explaining elements of his treatment led to a disorganized attempt to say how the interviewer has not experienced what it is like to be in a clinic or hospital as he has. He did not elaborate or finish this point, but despite the disorganized language, it is clear he is asking to be understood. 

The following is a more explicit example of a participant explaining the strong ties between diagnosis and social identity: 

FTD Participant: “like changing their schizophrenia paranoid, paranoid schizophrenia into the most dangerous mental illness there is. It was ranked so low on the, on the scale of mental illness that. It shouldn’t be considered that I’m dangerous or want to hurt someone or that I watch the children and I try to help them more than I help myself and the understanding I get from that is I got to stay thinking on myself. Not that so many differences … I talk in a language that try to be positive enough to for you to understand”.

Here, more than other quotes, we see how a mental illness identity is directly related to the negative stereotypes of being violent or predatory. After mentioning paranoid schizophrenia, he describes ways he may be viewed by others, ultimately bringing him to the conclusion he “got to stay thinking on myself”.

#### 3.1.2. Alienation Related to Social Experience

Alienation also appeared in narratives when participants described being rejected by their community or family. These descriptions often involved being “thought to be a problem” as in the following example:

FTD Participant: “And my schooling consisted of paddles. I always acted out and was thought to be a problem, because the nuns told me not to tell on myself or … let someone know that I feel different about life and the understand that the nuns gave me meant if you didn’t know it you wouldn’t grow it”.

Furthermore, another participant explained his family’s distaste for him and their misunderstanding of him. 

Non-FTD Participant: “I left [city], came back to [state] and um, that was a mistake. Cause I got locked up for child support. And then I got out and made child support for a while and then things didn’t go right. I run out of work or whatever. And um, I’ve always been in the doghouse with these people. Going all the way back to when I first started. So, it isn’t like they’re going to be ‘who is that guy?’ They’re going to know who I am”.

Here, the participant describes that he cannot live near family or else he will be prosecuted for child support, which he cannot pay due to his disability. This point was able to be derived, but his disorganization worsened as he continued to describe how his family views him. Both quotes illustrate a larger pattern where participants spoke about social groups, which then evolved into feeling unaccepted and unseen by others. Once negative affect was expressed, disorganization worsened. Of note, the second quote from a non-FTD participant was able to explain their stressful family dynamic to a greater extent prior to disorganization occurring.

Alienation was also the theme where differences between FTD and non-FTD groups were most apparent, both in frequency and quality. The FTD group had many more participants who described alienation. When alienation was mentioned, non-FTD participants typically had a greater capacity to elaborate on how the situation impacted them and how they made sense of it. Compare the following two illustrative quotes:

FTD Participant: “I was let down from my family, I believe. That’s what comes to my conclusion on the streets. In 1982 on the streets until 1996. The state had their realms around me, my freedom of choice was still in danger from the state until I took refuge at the [housing assistance] in 1998. Then my vocational rehabilitation and freedom of choice decided to incline”.

Non-FTD Participant: “Then in ’92 I hit the lottery. Boy, did my life change. It changed tremendously. I ended up losing my job over jealousy. It costs me money. The union people didn’t take care of me. They didn’t support me. I been working there 21 years. They said I missed three days and they were going to fire me. That’s what got to me. By me hitting the lottery I got jumped on, robbed and beat pretty bad”.

The FTD example is phrased from a more depersonalized, non-agentic perspective. Other than “I took refuge” and “I was let down from my family”, the participant speaks from the perspective of events happening to him. We see this most significantly when he stated “Then my vocational rehabilitation and freedom of choice decided to incline”. The FTD participant did not state why or what role he played in why he was let down by family, on the streets, or in danger from the state. In contrast, the non-FTD participant was able to explain how the lottery changed how he was treated at work and how he made sense of it (e.g., “losing my job over jealousy”, “that’s what got to me”). 

### 3.2. Interpersonal Tension

This theme highlighted participant reports of one-on-one social dynamics that produced negative affect and speech disorganization within the interview. Within this category, we identified two sub-themes: relationship tension and aggressive tension. 

#### 3.2.1. Relationship Tension

Here, participants described challenges in personal relationships, with their narratives often revealing vulnerabilities. This was most often seen in descriptions of past relationships, but in one case, also related to the current difficulties in addressing the interviewer during his narrative. “You’re a beautiful lady and I don’t know what it is, but I see some difference in you. It is, it’s more profitable than a lot of things. And you’re so adequate and proper. It would be hard to talk to you because I want to get your respect first (FTD participant)”. It is possible that his disorganization may be a product of feeling unable to authentically engage, which induced a sense of insecurity. 

Another participant displayed a similar pattern after describing a sexual experience in high school.

FTD Participant: “One day in home economics a little girl noticed I had a hard on. She tried to get up against me. I thought that can’t be done. I was afraid everybody was looking, the kids were looking. I wish things were different. Things weren’t different. I had psychological reality problems. Then in 12th grade by then I was getting good grades”.

He clearly describes his inner tension regarding his classmate’s proposition. As he elaborates on his feelings his speech becomes vague. “I wish things were different. Things weren’t different”. Disorganization further emerges when tangentially sharing his “psychological reality problems” and “good grades”.

#### 3.2.2. Aggressive Tension

This subtheme was less common and, when present, involved recollections of aggression and violence. Here, we see a participant’s description of prison become disorganized after expanding upon an experience of violence.

Non-FTD Participant: “Cause they know that there is two thirds black in there and one third white. The one third will get all the breaks and the two thirds will get all the screwing. And eh. One guard in particular was wanting to get it set up to where I was going to get jumped and have a tape player taken, a little tape player like that, some guy has something wrong with the door breaks, you know, where the door comes off, and eh, I would make little hinges from pieces of paperclip, melt them into the door and rehook the door back up”.

Much of the speech was fragmented, and disorganization emerged after describing discrimination within the prison and an instance of him being set up by a guard. It is possible topics inducing greater arousal, such as being in danger, produced negative affect, which expressed itself as disorganized storytelling. Within the interpersonal tension theme, this quote illustrates the pattern of non-FTD participants giving more thorough explanations of social challenges before disorganization emerges. Many FTD participants briefly touched on difficulties in their relationships before disorganization emerged, like the previous participants with their interviewer. Often, the non-FTD participants were able to withstand greater explanations of tension with others before disorganization emerged. For example, the previous non-FTD participant was able to explain the context of a racist environment and start explaining an act of violence towards him. The FTD participant only shared two sentences that described the actual interpersonal event.

### 3.3. Personal Benchmarks 

This theme focused on participants’ reflections on their personal aspirations, successes, and failures. It predominantly identified participants reflecting or sharing narrative experiences surrounding education, career, and personal fulfillment. Participants’ telling of personal benchmarks often developed into self-criticism or disappointment and disorganization, even if there were personal successes as well. We found two different processes in which topics led to affective reactivity: recognizing missed potential and inadequacy. 

#### 3.3.1. Recognizing Missed Potential

Content from this subtheme initially addressed hopes, desires, or past achievements, but often progressed into self-criticism, disappointment, or doubt. It was the only code that had both positive and negative affect words consistently preceding disorganization. Here, we see a participant explain contentment with his life, which progressed into elaborating on all he could not do.

FTD Participant: “I’ve done everything that I’ve wanted to, you know, except that it doesn’t cost a lot of money and I couldn’t afford it, of course I could have done something like that. But I’m not, I’m, like, you know, having trouble all over the world so I can’t say I’ve accomplish all my goals or all my dreams or desires. Um, I had a lot of jobs and I like dogs. Dogs are fun to play with and to uh, be your best friend”.

His speech became disorganized during and after critiquing his accomplishments and ultimately derailed. In another example, a participant described how he has persisted through hardship. This too initiates a shift into negative affect and disorganized content. 

Non-FTD Participant: “The only way to achieve good things, and have good things come from hard work and you can’t give up and you can’t use the word ‘I can’t do’ or ‘I can’t have’ or anything hard is something you don’t want to do. And uh, the last two months dealing with VA and what I been experiencing the last 6 months with the medication and not knowing that the medication could be the only thing that could haunt my life, is that I can see and feel right now”.

When comparing the two prior quotes, both reflected on the progress from the participant’s past. However, the non-FTD participant was able to provide more details on the distressing content prior to disorganization emerging.

#### 3.3.2. Inadequacy

This subtheme began with recollections of missed opportunities or regrets before leading to descriptions of insufficiency followed by disorganized speech. This subtheme differed from recognizing missed potential because it started directly from a missed opportunity or past regret. For example, this participant spoke about not receiving a sufficient education prior to devolving into disorganization and a potential persecutory delusion.

FTD Participant: “I could have went and got educated but I wanted to see the world. I went to Europe. It was different and they say they don’t consider prejudice in their uh, inauguration to the state, most don’t, there are a few that you have to watch, know the difference between how they think and what it is they know before they can get you in trouble”.

Although he did not specifically state that he regretted his lack of education, his narration quickly derailed into a fear of being targeted by European governments. Mentioning experiences that “could have” been may induce affective reactivity. 

More explicitly, in the following quote, a participant states feeling regret about not seeking treatment earlier and progresses into disorganization.

Non-FTD Participant: “I hid it for so long. I hid I had a problem. This isn’t the type of problem you talk to anybody about. I hide it for 10 years. My mother told me that if I don’t get help that she was going to put me out. So I got help. Now I’m in a program to make me better, which I should have been in a long time ago. That’s about it. That pretty well sums it up. Kids in school then I jump in here. My childhood was alright. You know what I’m saying. I was in the groove of things”.

The non-FTD participant was able to share vulnerably about hiding his mental health before significant disorganization emerged. On the other hand, the FTD participant briefly mentioned his lack of education before disorganization appeared. 

### 3.4. Adverse Experience

Adverse experience is the final category of themes we found to precede disorganization. This code involved self-reflection of past adverse experiences without a significant social component. It differs from personal benchmarks in that it addressed the most seemingly challenging or painful events shared in their narrative interview. We found that these topics could progress into disorganization without or with an overt emotional process, which led to us separating the categories into two subthemes: past hardship and grief. 

#### 3.4.1. Past Hardship

This subtheme involved narratives of difficult experiences without a clearly identifiable emotional process. It seemed like there was an effort for participants to remain objective, detached, and conservative in their explanations. 

Here, we see one of the longest quotes within this study. His narrative was linear in describing his movement into foster care, the death of his mother, and daily activities in the children’s home. That said, he dipped in and out of disorganization throughout the quote as difficult experiences were described as follows:

FTD Participant: “I think I was 12. I was taken to [children’s home]. I still had no freedom of choice. As time passed by, my mom died when I was 13 years old. I never got to see my mom. She tell me what she wanted me to do in my life. My sisters wouldn’t take me to see her on her hospital bed before she died. And that hurt me more. At that time my brothers, all three of my brothers, [name], [name], and [name]. were in the military. At that time, that was back in 1967. They got an emergency leave because of my mom’s funeral. Still I was disgusted and didn’t have the freedom of choice. It’s been a little bit harder. At the time of the children’s home, I was working around the home, raking leaves, mowing grass, picking weeds from the garden that they had at the [name] children’s home, and then going to school. And then cleaning up the building at the time. I have left out a few other details. It’s been a little bit harder. At the end of that time, after 2 years, I was 14, I came out of [name] children’s home with a standard of honorable, I was an honor with a merit of disciplinary action. I succeeded and continued to be an honor in my disciplinary action”.

We see the first signs of disorganization through the repetition of “freedom of choice”. This was an undefined term that he repeated several more times throughout his entire narrative interview. Here, we see this term occur after two painful experiences: being surrendered to foster care and not being able to see his mother before she died. Disorganization continues to emerge as he makes an unclear reference when stating “It’s been a little harder”. This is another phrase he repeats later. He then repeats phrases of “honor” and “disciplinary action” after describing his daily activities at the children’s home. This disorganization may be a sign of his time at the children’s home being distressing as well. The participant disclosed limited emotions regarding objectively painful experiences. It is important to note how disorganization occurred after certain events he described. This was a shared theme between several other participants after describing an adverse event without divulging their personal feelings. 

#### 3.4.2. Grief

The grief subtheme consisted of difficult narrative experiences with an overt emotional component. These emotionally charged narratives often led to increased expressions of negative affect prior to transitioning into disorganized speech. For example, this participant explained a prior emotional experience, which may cause the participant to feel an echo of negative affect capable of inducing affective reactivity.

Non-FTD Participant: “I couldn’t realize what I was going through because of that money. I lose my job, my wife, I lose everything. Today I think about all that stuff, I think about bad things I want to go do. That part came on, my life be sitting at home, acting scared. I’m here now in this office”.

As we see in this quote, discussion of past loss and grief leads straight into what his present concerns are and reflection of how his losses have accumulated to being “here now in this office”. Disorganization clouds this process of connecting past grief to the present moment with an unclear timeline, missing context, and vague references.

Within the adverse experience theme, we see disorganization emerge after more intimate details of distressing content in the non-FTD population. The non-FTD participant can clearly say how winning the lottery negatively impacted him, whereas the FTD participant mentioned few difficulties from their children’s home prior to disorganization emerging. 

## 4. Discussion

Although prior research has shown relationships between affective reactivity and disorganized symptoms, the current study moves beyond traditional quantitative analyses by allowing an in-depth understanding of how specific topics lead to disorganization. In this study, we identified four primary themes (alienation, interpersonal tension, personal benchmarks, adverse experiences) that consistently preceded disorganization. Each theme was further subdivided into two distinct subthemes for a more nuanced characterization of the content preceding disorganized speech. In addition, our team also separated those with schizophrenia into closely matched FTD and non-FTD subgroups, allowing us to observe similarities and key differences in content that led to disorganization.

Our findings support existing literature that links disorganized speech with a host of psychological factors, including social cognition [39,40,41], metacognitive processes [42,43,44], and traumatic experiences [45,46]. This suggests that disorganized speech is not meaningless content but that it may emerge in response to the multifaceted experiences within the lives of those with schizophrenia. This perspective aligns with stress reactivity studies [47,48,49] and the work of Searles [50] and subsequent researchers [20,21], who have observed that disorganized speech can be a protective mechanism for psychological distress. Although prior case studies have hinted at personal meaning prior to disorganization [19,20,21,22], our study is the first study to explicitly examine precursors of disorganized speech in interviews with a group of people with schizophrenia. We found that narrative elements leading to negative affect are important precursors to disorganization. This finding adds to quantitative research, which has identified negative affect as an antecedent to disorganization [3,6].

The theme of alienation was particularly noteworthy. Participants frequently transitioned into disorganized speech when discussing experiences of being misunderstood or ostracized based on their mental illness or within social contexts. This underscores the emotional impact of alienation on patients and aligns with literature highlighting the influence of social alienation on the mental health of those with psychotic disorders [51,52]. It also may reflect the importance of stigma, as those who exhibit more psychotic symptoms are often both more alienated and stigmatized. Notably, the alienation category also contained the largest discrepancies when comparing FTD (a highly stigmatized group) and non-FTD participants. It is possible those with FTD exhibited a higher frequency of alienation codes preceding disorganization because they experience more alienation—both socially and due to the severity of their illness—than those without FTD. This observation suggests that alienation, especially in the context of FTD, could serve as a major barrier to recovery and should be a key focus of therapeutic interventions.

Our findings across themes often supported disorganization as a negative affective reaction to stress, and the literature pointed to this reaction as a protective mechanism to avoid feeling psychological pain. Regarding stress reactivity, there is a robust body of literature that demonstrates the role of stress as a risk factor for psychotic episodes [49] and the ability to return to baseline after a stressful event as a possible sign of resilience in those with psychosis [47]. Those with FTD may be particularly vulnerable to affective stress [18]. The theme of interpersonal tension underscored the sensitivity of individuals with schizophrenia to social dynamics. Challenges within relationships appear to evoke stress and possible feelings of insecurity, subsequently leading to disorganization. This finding that discussing personal relationships may induce disorganized speech has implications for therapeutic interventions, as it highlights the need for strategies to navigate this tension. Similarly, the theme of personal benchmarks reveals a pattern where self-assessment potentially led to emotions triggering disorganized speech. This understanding could be vital for therapists, as it suggests the importance of carefully guiding clients through self-assessment without assuming past achievements will lead to positive feelings—they may instead induce stress, which could trigger disorganization.

In addition to stress reactivity, there is also the idea that disorganization could be a protective mechanism to avoid confronting painful experiences. These ideas have been present since Searles [50] observed that psychotic content emerges in response to intolerable pain in one’s sense of self. In this study, the theme of adverse experiences also revealed that discussing past hardships and grief can lead to disorganized speech. This aligns with prior research claiming that disorganized speech can manifest as a coping mechanism for unresolved trauma, protecting clients from feeling overwhelming psychological pain [53,54]. Recent case studies have reinforced this perspective by finding disorganization to emerge from psychologically painful areas [20,21]. After speaking about the function of disorganization with their client, de Jong and colleagues [19] paraphrased the participant in their case study, saying “children flee into fairytales and that he himself fled into psychosis”. The current study further documents disorganization as a response to affectively negative and painful topics across 30 participants outside of psychotherapy. In conjunction with research on affective reactivity [6,15,16], disorganization may function as a protective mechanism for many individuals with psychosis.

Future directions in this line of research could explore how psychotherapy can best build clients’ awareness of unique triggers for disorganization. Given the matched pairs design of this study (FTD and non-FTD groups), we were able to yield insights specific to those with high and low levels of disorganization. Our study suggests that as disorganized symptoms become more pronounced, triggers exacerbating disorganized speech may increasingly center on instances of alienation. As a result, it may be more effective for psychotherapy to initially focus on building awareness of alienation triggers in highly disorganized clients. Furthermore, descriptions of illness experience seem to trigger disorganization almost exclusively in FTD participants. Thus, it is plausible that feelings of being misunderstood due to their diagnosis worsen when disorganized symptoms are more severe. Future research could determine if psychoeducation and meaning making around one’s illness can mitigate stigma and disorganization. When FTD and non-FTD groups were compared across themes, more intimate and detailed descriptions of distressing topics were present prior to disorganization emerging in the non-FTD group. This could be seen across several codes but was clearest in the alienation codes, which were more frequently observed in the FTD group. This could signal the importance of focusing on feelings of alienation in interventions seeking to treat FTD.

This study has limitations. All qualitative analyses were conducted with typed narrative interview transcripts. As a result, factors such as non-verbal communication, tone of voice, and pace were not considered. Access to audio or video recordings of the interviews could provide a more nuanced understanding of disorganization and emotional processes. Although richer findings may have been attainable with multimodal data, it is important to highlight that this study sought to mitigate these limitations. We employed two measures of disorganization and held consensus meetings among raters. Additionally, we employed consensus procedures between four qualitative coders to mitigate assumptions about participants’ emotional processes. The entirely male sample is also a limitation that could impact the generalizability of these findings. We only included males in this study because the parent study sample was predominantly male, and it allowed us to match participants on core demographic features. Another limitation is centered around the use of the IPII. The open-ended nature of the IPII is one of the reasons it was chosen; it allows participants the freedom to control the topics being discussed. However, the lack of follow-up questions from therapists may also impact generalizability as this is a different process than a typical social interaction or therapy session.

## 5. Conclusions

In summary, this study provides a nuanced qualitative exploration into the precursors of disorganized speech in individuals with schizophrenia, highlighting themes like alienation, interpersonal tension, personal benchmarks, and adverse experiences. These findings underscore the complex interplay between personal experiences and disorganized speech, offering valuable insights for therapeutic interventions. Particularly, this study suggests the need for tailored approaches in psychotherapy, focusing on individual triggers and emotional processes. Future research should consider the integration of multimodal data, including non-verbal cues, to further enrich our understanding of disorganization in schizophrenia. This study marks a significant step towards a more empathetic and personalized understanding of schizophrenia, paving the way for more effective, targeted interventions to aid in recovery.

## Data Availability

Data from this project are available upon request.

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
