# Peer review of "Narrative Forewarnings: A Qualitative Analysis of the Themes Preceding Disorganized Speech in Schizophrenia"

_behavsci, 2024, doi:10.3390/bs14030212_

Round 1

Reviewer 1 Report

Comments and Suggestions for Authors

Thank you for the opportunity to review interesting paper on the use of narrative to understand precursors of disorganized speech among people with schizophrenia. I commend the authors for their attention toward using rich, narrative-derived information to understand qualitative aspects of disorganization, as this is an important and largely understudied area of research. Overall the manuscript is well-written, with excellent detail in the methods regarding the coding of the themes from the narratives. At the same time, I have questions about the current draft that should be considered and/or addressed as the authors move forward with this interesting work.

Overall Comments:

1.      Paper is very well-written and organized.

2.      I appreciate the level of detail given in the Method re: development of codebook, consensus meetings, determination of saturation, etc. This made it very easy to understand what the authors actually did.

3.     Excellent rationale in the intro for the clinical relevance of this paper, which I believe is important for this journal and this special issue.

4.      Inclusion of direct statements from narratives is great for contextualizing themes.

Questions:

1.     In the Method section for the Narratives, it reads that the data from this study were drawn from the initial IPII prompt to tell one’s life story. Were the data limited to that part of the IPII? If yes, what is the rationale for excluding the remaining portions of the interview discussing insight, impact on function, perception of future, etc?

2. The authors note several times in the paper that their data supported “disorganization as a protective mechanism to avoid feeling psychological pain.” This is an interesting idea but is under-developed in the paper, both in the introduction and in how it is reviewed in the discussion. For example, is it not a more parsimonious explanation that these results are explained only by stress reactivity?

3. The use of the split PANSS P2 approach for determining FTD and non-FTD might warrant more consideration/explanation. Can the authors provide any citations or precedent for this approach? Overall the PANSS was not designed to be used this way, but might be ok if more explanation can be provided. Might also be worth reporting P2 means and s.d. for each group to understand how scores varied. Additionally, the authors have CDI scores in this study. The CDI is arguably a much more comprehensive tool for assessing the presence of FTD relative to PANSS P2. Did the authors consider using CDI scores to group FTD vs. non-FTD participants instead? Depending on the distribution of the CDI data, could even consider a median split to make things a bit more data-driven.

Author Response

Please see the attachment for responses to all points raised by reviewers.

Reviewer 2 Report

Comments and Suggestions for Authors

Author Response

(The authors gave the same response as above.)

Round 2

Reviewer 2 Report

Comments and Suggestions for Authors The changes seems satisfactory and I recommend publication.